# Advancing LLM Reasoning Generalists with Preference Trees

**Lifan Yuan** [* 1 2]  **Ganqu Cui** [* 1]  **Hanbin Wang** [* 3 4]  **Ning Ding** [1]  **Xingyao Wang** [2]  **Jia Deng** [5]  **Boji Shan** [6]
**Huimin Chen** [1]  **Ruobing Xie** [7]  **Yankai Lin** [5]  **Zhenghao Liu** [3]  **Bowen Zhou** [1]  **Hao Peng** [2]  **Zhiyuan Liu** [1]
**Maosong Sun** [1]

## Abstract

We introduce EURUS, a suite of large language models (LLMs) optimized for reasoning. Fine-tuned from Mistral-7B and CodeLlama-70B, EU-RUS models achieve state-of-the-art results among open-source models on a diverse set of benchmarks covering mathematics, code generation, and logical reasoning problems. Notably, EURUS-70B beats GPT-3.5 Turbo in reasoning through a comprehensive benchmarking across 12 tests covering five tasks, and achieves a 33.3% pass@1 accuracy on LeetCode and 32.6% on TheoremQA, two challenging benchmarks, substantially outperforming existing open-source models by margins more than 13.3%. The strong performance of EU-RUS can be primarily attributed to **ULTRAINTER-ACT**, our newly-curated large-scale, high-quality alignment dataset specifically designed for complex reasoning tasks. ULTRAINTERACT can be used in both supervised fine-tuning and preference learning. For each instruction, it includes a preference tree consisting of (1) reasoning chains with diverse planning strategies in a unified format, (2) multi-turn interaction trajectories with the environment and the critique, and (3) pairwise data to facilitate preference learning. UL-TRAINTERACT allows us to conduct an in-depth exploration of preference learning for reasoning tasks. Our investigation reveals that some well-established preference learning algorithms may be less suitable for reasoning tasks compared to their effectiveness in general conversations. In-

spired by this, we derive a novel reward modeling objective which, together with ULTRAINTERACT, leads to a strong reward model. Models and data will be made public.

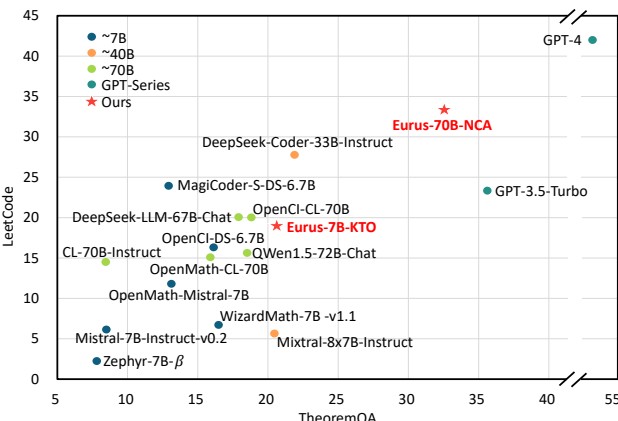

*Figure 1.* Evaluation results on LeetCode and TheoremQA, two challenging OOD coding and math benchmarks with only test sets. Our EURUS-7B is comparable with baselines that are 10x larger and EURUS-70B is the only one on par with GPT-3.5 Turbo.

## 1. Introduction

Current alignment techniques have significantly advanced the development of open-source large language models (LLMs) that effectively meet user expectations and align with human values (Touvron et al., 2023; Tunstall et al., 2023). On complex reasoning, success has been achieved by specializing models for specific capabilities, such as coding (Wei et al., 2023; Guo et al., 2024a; Zheng et al., 2024) and solving math problems (Fu et al., 2023; Yue et al., 2023; Luo et al., 2023a; Toshniwal et al., 2024). However, these models still fall short, by large margins, of the most advanced proprietary models in their all-around capabilities to tackle a diverse range of challenging problems. We conjecture that this performance gap can be primarily attributed to (1) the lack of high-quality alignment data and (2) the under-exploration of preference learning techniques for improving models' complex reasoning capabilities. In this paper, we take strides towards bridging this gap by addressing both factors and developing EURUS.

---
[*]Equal contribution  [1]NLP Group, DCST, IAI, BN-RIST, Tsinghua University  [2]University of Illinois Urbana-Champaign  [3]Northeastern University  [4]ModelBest.Inc  [5]Renmin University of China  [6]BUPT  [7]Tencent. Correspondence to: Lifan Yuan <lifan4@illinois.edu>, Ganqu Cui <cgq22@mails.tsinghua.edu.cn>, Hanbin Wang <wanghanbin-panda@gmail.com>, Ning Ding <dn97@mail.tsinghua.edu.cn>, Zhiyuan Liu <liuzy@tsinghua.edu.cn>.

*The first AI for MATH Workshop at the 41st International Conference on Machine Learning*, Vienna, Austria. Copyright 2024 by the author(s).

EURUS consists of a suite of LLMs finetuned from Mistral-7B (Jiang et al., 2023a) and CodeLLaMA-70B (Roziere et al., 2023). Across a diverse set of complex reasoning benchmarks that are mostly out-of-distribution (OOD), EU-RUS achieves state-of-the-art overall performance among all open-source models. In particular, EURUS excels in solving challenging problems that often require sophisticated planning, reasoning, tool integration, and the ability to interact with and learn from the environment and users. As shown in Figure 1, on university-level STEM questions TheoremQA (Chen et al., 2023) and competition-level coding problems LeetCode Contest (Guo et al., 2024a), EU-RUS-70B significantly outperforms all open-source models, achieving comparable performance to GPT-3.5 Turbo.

EURUS models are trained on ULTRAINTERACT, our newly-curated, large-scale, and high-quality alignment data specifically designed to improve LLMs' reasoning capabilities. ULTRAINTERACT consists of a diverse set of instructions spanning math, coding, and logical reasoning problems from 12 established datasets. For each instruction, ULTRAINTERACT collects a preference tree that includes: (1) **Diverse planning strategies in a unified pattern,** such as sequential processing (Wei et al., 2022) and tool creation (Qian et al., 2023), followed by executing step-by-step actions formatted in either text or code, to provide divserse reasoning trajectories. (2) **Multi-turn interaction trajectories with the environment and the critique,** to improve models' capabilities to learn from feedback and correct previous errors (Wang et al., 2023b). (3) **Paired correct and incorrect actions organized in tree structures,** to facilitate preference learning. In total, ULTRAINTERACT contains 86K instructions and 220K action pairs, where each pair consists of an instruction, a correct response, and an incorrect one. Conceptually, ULTRAINTERACT's data resemble imbalanced binary trees as shown in Figure 2.

ULTRAINTERACT can be used in both supervised fine-tuning and preference learning. Our experiments show that, using ULTRAINTERACT along with established datasets in instruction fine-tuning already achieves strong performance. ULTRAINTERACT further facilitates preference learning for reasoning tasks, improving the performance even further with KTO (Ethayarajh et al., 2024) and NCA (Chen et al., 2024a). Surprisingly, applied to an instruction finetuned EURUS model, DPO (Rafailov et al., 2023) hurts the performance.

Through careful analysis, we provide evidence that the performance in reasoning correlates with the value of rewards of chosen data—a higher final reward often indicates a better reasoning capability. Besides, our investigation suggests that DPO may be less suitable for reasoning tasks than KTO and NCA. Inspired by this fresh finding, we devise a new objective for reward modeling to augment the Bradley-Terry

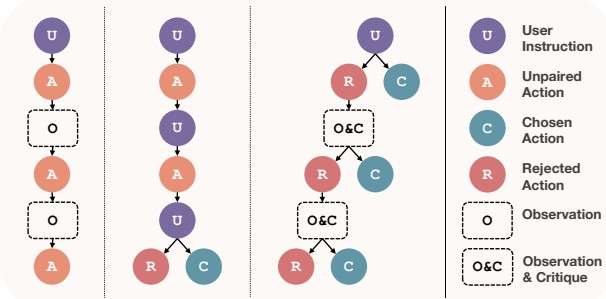

*Figure 2.* Left: CodeActInstruct (Wang et al., 2024) and Code-Feedback (Zheng et al., 2024); Middle: HH-RLHF (Bai et al., 2022); Right: ULTRAINTERACT. Each instruction in ULTRAIN-TERACT is constructed as a preference tree.

objective (Bradley & Terry, 1952), explicitly encouraging training to increase the absolute rewards of chosen solution and decrease those of rejected data. Furthermore, UL-TRAINTERACT leads to our reward model EURUS-RM-7B, which achieves a better correlation with human annotators than all existing models on AutoJ (Li et al., 2023a) and MT-Bench (Zheng et al., 2023), including GPT-4 (OpenAI, 2023). EURUS-RM-7B demonstrates especially strong preference modeling performance on reasoning tasks.

## 2. ULTRAINTERACT: Tree-structured Alignment Data for Reasoning

Solving complex problems often requires the model's capability in planning and reasoning, integrating with tools, and interacting with and learning from both the environment and the users. This is reflected in ULTRAINTERACT's design choices: (1) Its instructions are diverse, challenging, and of a large scale (§2.1); (2) It provides multi-turn trajectories that solve the input instruction through multiple turns of interaction with and learning from the environment and critique. At each turn, it breaks down the problem into smaller ones (§2.2). (3) ULTRAINTERACT includes pairwise data to facilitate preference learning (§2.3).

Conceptually, ULTRAINTERACT collects a preference tree for each instruction, with the instruction being the root and each action a node (Fig. 2). A trajectory is a root-to-leaf path consisting of a sequence of actions. In each preference tree, all nodes of correct actions and all trajectories ending with correct actions can be used for SFT. Paired correct and incorrect nodes or trajectories can be used for preference learning.

### 2.1. Instruction Selection Emphasizing Complexity, Quality, and Diversity

We target three representative reasoning tasks: math problem-solving, code generation, and logical reasoning. The complexity, quality, and diversity of the alignment data are crucial to the model's performance (Liu et al., 2023). Fol-

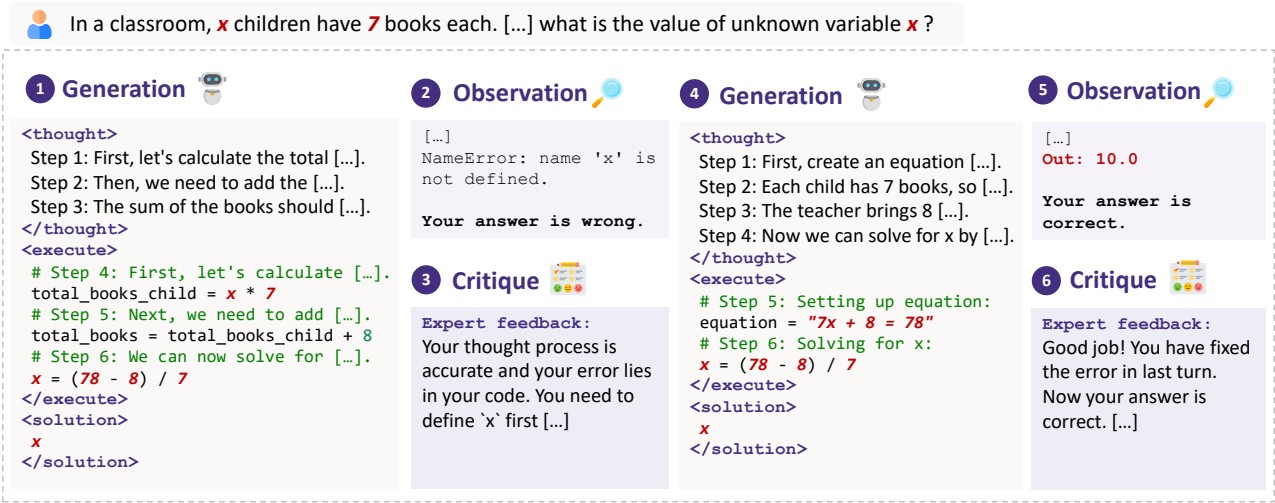

*Figure 3.* An illustrative example of an ULTRAINTERACT trajectory over two turns. In each turn, the actor model generates step-by-step reasoning chains, and the environment and the critique model provide observations and textual critique respectively.

lowing Wang et al. (2023b), we select challenging problems that GPT-3.5-Turbo fails to solve. We intentionally restrict the selection of the datasets to those with ground-truth solutions, aiming to ensure high-quality oversight signals rather than relying on LLM-as-a-judge annotation (Weyssow et al., 2024). Besides, the gold solutions also serve as references for the critique model to generate feedback. To promote ULTRAINTERACT's diversity, we pick datasets of different categories. For each dataset, we include distinct reasoning patterns based on question categories or formulations necessary to solve the problems. Table 6 summarizes the datasets selected by ULTRAINTERACT. Except for MATH, none of the training datasets is used in our evaluation.

### 2.2. Decomposition and Interaction at Each Turn

Figure 3 provides an illustrative example. In what follows, we connect the actor model with a Python interpreter as the "environment". Unless otherwise specified, we use GPT-3.5 Turbo as the actor model.

Following Wang et al. (2024), the actor model first decomposes the input problem into several sub-problems and then solves each by generating Python code pieces as actions and using the environment to execute them. To promote solution diversity, the actor model randomly samples one reasoning schema in the form of either CoT (Wei et al., 2022) or modularization programming (Qian et al., 2023; Yuan et al., 2023). The actor then generates actions in text or code to solve each sub-problem, with each step being marked by explicit notations.

Multi-turn interactions with the environment are often necessary to solve challenging problems (Wang et al., 2023b).

To improve such capabilities of the models, ULTRAINTER­ACT collects trajectories in which the actor model interacts with the environment and a critique model (a proxy for user) and refines its action based on their feedback.

The environment receives an action from the actor model along with the interaction history, and then the code interpreter returns two kinds of "*Observation*": (1) Python execution results, either program outputs or error traceback messages; (2) binary feedback, indicating whether the solution is correct or not. Then, the observations along with the history will be passed to a *critique* model, which locates the errors and provides suggestions for improvements. To avoid potential bias introduced by self-correction (Wang et al., 2023b; Xu et al., 2024), we adopt a stronger model, GPT-4, as the critique and ensure critique quality by providing GPT-4 with ground truth answers as references.

This procedure resembles Wang et al. (2024). However, we adopt more diverse reasoning patterns to teach LLMs to learn rationales rather than simply memorizing answers (Mitra et al., 2023), and learn to create and use tools (Qian et al., 2023; Yuan et al., 2023; Qin et al., 2023). Besides, we believe that it is important for LLMs to learn from the feedback provided by the critique rather than solely from observations of the environment.

### 2.3. Preference Trees Facilitates Preference Learning Across Multiple Turns

Unlike open-ended conversations, where human preference is ambiguous and challenging to specify, many reasoning tasks have clear and objective preferences for *correct* actions. The preference annotation is threfore an evaluation

*Table 1.* Some statistics of ULTRAINTERACT.

| Task | Type | | # Instructions | # Turns per Traj. | | | | | # Tokens per Traj. | Avg. # Traj per Ins. | Total # Pairs | # Correct Answers |
|------|------|------|------|------|------|------|------|------|------|------|------|------|
| | w/ Interaction? | w/ Tool? | | T1 | T2 | T3 | T4 | T5 | | | | |
| Math | ✓ | ✓ | 22,928 | 10,440 | 4,122 | 1,898 | 904 | 5,564 | 1,750.0 | 1.0 | 42,780 | 68,033 |
| | ✗ | ✓ | 2,757 | 16,154 | - | - | - | - | 439.1 | 5.9 | 13,217 | 16,154 |
| | ✓ | ✗ | 22,639 | 10,708 | 3,521 | 1,459 | 723 | 6,228 | 1,521.9 | 1.0 | 44,750 | 62,182 |
| | ✗ | ✗ | 2,083 | 16,348 | - | - | - | - | 538.1 | 7.8 | 12,624 | 16,348 |
| Coding | ✓ | - | 20,463 | 13,265 | 2,584 | 987 | 379 | 3,248 | 1,728.5 | 1.0 | 18,106 | 22,215 |
| | ✗ | - | 8,495 | 92,618 | - | - | - | - | 1,070.4 | 5.5 | 78,634 | 92,618 |
| Logic | ✓ | ✓ | 2,086 | 1,685 | 298 | 72 | 8 | 23 | 1,299.8 | 1.0 | 1,750 | 2,198 |
| | ✓ | ✗ | 4,467 | 2,453 | 1,674 | 340 | 0 | 0 | 1,266.7 | 1.0 | 7,958 | 7,231 |
| Total | - | - | 85,918 | 163,671 | 12,199 | 4,756 | 2,014 | 15,063 | 1,201.8 | 2.3 | 219,819 | 286,979 |

of the correctness of the solutions *conditioning ground truth ones*, which come with the datasets in ULTRAINTERACT. This eliminates the need for human or LLM-based preference annotation and ensures high data quality. To facilitate preference learning, ULTRAINTERACT pairs correct and incorrect actions.

**Sampling Paired Correct and Incorrect Actions at Each Turn.** For each instruction in ULTRAINTERACT, we sample, from the actor model, a pair of correct and incorrect actions following §2.2. We follow (Cui et al., 2023) to sample the pair from different actor models to ensure response diversity. To prevent models from exploiting shortcuts based on surface features, we exclude instances that fail to pass the Python syntax check.

Certain challenging problems in ULTRAINTERACT pose difficulties in obtaining correct actions, even using strong actors such as GPT-4, with nearly zero *pass@100* accuracies. To improve the pass rates of the actor models while keeping the expense under control, we sequentially take the following steps. (1) Directly sampling 20 actions and randomly keeping a correct one, if any. (2) If no correct action is obtained, we repeat the above process up to three times, progressively switching from more cost-effective models to the strong yet expensive GPT-4 Turbo. (3) For the remaining difficult problems where no correct action is acquired after the previous two steps, we provide the actor with ground-truth rationales and answers, and then apply various techniques to elicit correct actions. The specific information provided and the techniques applied vary depending on the tasks (Appendix A.2).

**Tree-structured Action Pairs Across Multiple Turns.** After each turn, the correct action concludes its trajectory. We expand the *incorrect* action into the next turn, and have the actor interact with the environment and the critique to refine its solution (§2.2). We then repeat the procedures introduced earlier in this section to collect an additional action pair. By expanding the *incorrect* action, ULTRAINTERACT can provide data to help models

learn from feedback, and collect multiple action pairs for preference learning across multiple turns.

Conceptually, for every instruction, ULTRAINTERACT constructs a binary preference tree with each action being a node (Figure 2). We cap the tree at a maximum of five turns.

**Additional Instruction-action Pairs for Challenging Problems.** We believe the challenging instructions that make it to step (3) above can provide valuable training signals. Therefore, for a subset of these problems with multiple ground truth solutions, we further sample additional correct actions to cover all ground truths. Accordingly, we further sample incorrect actions to pair with these additional correct actions, so that they can be used in both supervised fine-tuning and preference learning.

With the tree-structured data, ULTRAINTERACT enables comparisons at every turn, in contrast to comparing only at the last turn (Bai et al., 2022), and thus can improve the models' interaction ability. Closing this section, Table 1 summarizes some statistics of ULTRAINTERACT, and more details are in Appendix A.4.

## 3. EURUS: State-of-the-art Open LLMs in Reasoning

ULTRAINTERACT helps us develop EURUS, a suite of LLMs and a reward model (RM).

**Supervised Fine-Tuning.** EURUS-7B-SFT is fine-tuned from Mistral-7B (Jiang et al., 2023a) and EURUS-70B-SFT from CodeLLaMA-70B (Roziere et al., 2023). First, we perform SFT using all correct actions (287K) in ULTRAINTERACT. We find it yields better performance to discard interaction history and train only on correct leaf nodes in each tree. To improve general instruction-following ability, we include into our SFT data mixture UltraChat (Ding et al., 2023), ShareGPT[1], and OpenOrca (Lian et al.,

---

[1] https://huggingface.co/datasets/openchat/openchat_sharegpt4_dataset

*Table 2.* Open-source LLM baselines that we compare to.

| Type | Models |
|---|---|
| **General Purpose** | Mistral-7B-Instruct-v0.2 (Jiang et al., 2023a), Zephyr-7B-$\beta$ (Tunstall et al., 2023), OpenChat-3.5-1210 (Wang et al., 2023a), Starling-LM-7B-$\alpha$ (Zhu et al., 2023), Mixtral-8x7B-Instruct (Jiang et al., 2023a), DeepSeek-LLM-67B-Chat (DeepSeek-AI, 2024), QWen1.5-72B-Chat (Bai et al., 2023) |
| **Coding** | Magicoder-S-DS-6.7B (Wei et al., 2023), OpenCodeInterpreter (OpenCI for short, DS-6.7B/CL-70B) (Zheng et al., 2024), DeepSeek-Coder-33B-Instruct (Guo et al., 2024a), and CodeLLaMA-70B-Instruct(Roziere et al., 2023). |
| **Math** | MAmmoTH-7B-Mistral (Yue et al., 2023), WizardMath-7B-v1.1 (Luo et al., 2023a), OpenMath (Mistral-7B/CodeLLaMA-70B) (Toshniwal et al., 2024). |

*Table 3.* Overall performance. All test sets except MATH are out-of-distribution to our models and most baselines. MAmmoTH, OpenChat, and Starling-LM have been trained on TheoremQA test sets. We ~~strikethrough~~ the contaminated numbers.

| Model | Coding | | | Math | | | | | Reasoning | Ins-Following | Multi-Turn | | Avg. |
|---|---|---|---|---|---|---|---|---|---|---|---|---|---|
| | HumanE. | MBPP | LeetC. | GSM-Plus | MATH | Theo.QA | SVAMP | ASDiv | BBH | IFEval | Code | Math | |
| ~7B | | | | | | | | | | | | | |
| Mistral-7B-Instruct-v0.2 | 39.0 | 30.8 | 6.1 | 15.7 | 9.5 | 8.5 | 42.9 | 49.5 | 62.4 | 44.4 | 7.4 | 26.2 | 28.5 |
| Zephyr-7B-$\beta$ | 29.3 | 35.8 | 2.2 | 23.3 | 5.0 | 7.8 | 19.1 | 28.0 | 61.8 | 39.7 | 5.2 | 16.9 | 22.8 |
| OpenChat-3.5-1210 | 64.0 | 61.7 | 11.7 | 46.7 | 28.1 | ~~19.1~~ | 75.4 | 77.0 | 67.0 | **50.3** | 21.3 | 32.4 | 46.2 |
| Starling-LM-7B-$\alpha$ | 46.3 | 51.1 | 8.9 | 23.7 | 21.5 | ~~12.0~~ | 26.3 | 39.8 | 67.1 | 26.1 | 18.4 | 28.9 | 30.8 |
| Magicoder-S-DS-6.7B | 75.6 | **70.4** | **23.9** | 16.4 | 19.9 | 13.1 | 61.6 | 62.8 | 57.0 | 21.1 | **27.9** | 8.0 | 38.1 |
| OpenCI-DS-6.7B | **76.8** | 66.2 | 16.1 | 41.5 | 31.6 | 16.1 | 74.5 | 79.8 | 53.9 | 22.6 | 5.9 | 1.3 | 40.5 |
| MAmmoTH-7B-Mistral | 24.4 | 42.4 | 7.2 | 40.1 | 36.0 | ~~26.3~~ | 60.7 | 72.3 | 57.7 | 34.9 | 3.7 | 6.7 | 34.4 |
| WizardMath-7B-v1.1 | 50.0 | 53.9 | 6.7 | 54.6 | 30.0 | 16.5 | 57.8 | 73.5 | 64.4 | 22.6 | 16.2 | 8.9 | 37.9 |
| OpenMath-Mistral-7B | 33.5 | 46.6 | 11.7 | **59.4** | **39.1** | 13.1 | 83.4 | 79.8 | 58.6 | 15.0 | 2.9 | 5.3 | 37.4 |
| EURUS-7B-SFT | 55.5 | 59.1 | 20.0 | 52.1 | 32.6 | 20.0 | 82.2 | 84.1 | 64.6 | 44.0 | 15.4 | 28.4 | 46.5 |
| + DPO | 50.6 | 52.1 | 8.3 | 51.0 | 28.3 | 20.9 | 78.7 | 83.8 | 65.0 | 42.5 | 20.6 | 32.4 | 44.5 |
| + KTO | 56.1 | 58.6 | 18.9 | 55.0 | 33.2 | 20.6 | 84.4 | 85.0 | **67.6** | 43.1 | 19.1 | **43.6** | **48.8** |
| + NCA | 55.5 | 60.2 | 14.4 | 54.9 | 34.2 | **20.9** | **84.6** | **85.4** | 64.3 | 42.7 | 21.3 | 38.7 | 48.1 |
| ~40B | | | | | | | | | | | | | |
| Mixtral-8x7B-Instruct | 50.6 | 50.1 | 5.6 | **49.6** | 25.9 | 20.4 | 66.4 | 68.8 | **73.5** | **48.8** | 12.5 | **37.3** | 42.5 |
| DeepSeek-Coder-33B-Ins | **82.3** | **73.9** | **27.8** | 29.5 | 20.2 | **21.9** | 75.2 | **85.0** | 61.5 | 26.1 | **35.3** | 21.8 | **46.7** |
| ~70B | | | | | | | | | | | | | |
| CodeLLaMA-70B-Instruct | 56.7 | 58.6 | 14.4 | 34.9 | 12.0 | 8.4 | 63.5 | 70.1 | 74.5 | 24.0 | 3.7 | 14.2 | 36.3 |
| DeepSeek-LM-67B-Chat | 70.7 | 65.7 | 20.0 | 65.0 | 41.0 | 17.9 | 74.0 | 84.0 | 78.9 | 52.7 | 30.9 | 41.8 | 53.5 |
| QWen1.5-72B-Chat | 71.3 | 56.9 | 15.6 | **65.4** | 43.4 | 18.5 | 79.5 | 79.1 | 78.0 | **53.4** | 27.2 | 38.2 | 52.2 |
| OpenCI-CL-70B | 77.4 | 71.7 | 20.0 | 46.1 | 29.2 | 18.8 | 76.1 | 79.4 | 66.7 | 26.8 | 30.9 | 12.0 | 46.3 |
| OpenMath-CL-70B | 39.0 | 52.6 | 15.0 | 62.2 | **45.9** | 15.9 | 86.6 | 82.8 | 59.9 | 15.7 | 14.0 | 0.4 | 40.8 |
| EURUS-70B-SFT | 75.6 | **74.2** | 33.3 | 58.1 | 40.6 | 28.0 | 86.3 | 88.5 | 79.9 | 49.2 | 31.6 | 40.4 | 57.1 |
| + KTO | 76.8 | 68.2 | 26.1 | 62.2 | 41.3 | 30.6 | **90.4** | 89.0 | **80.8** | 46.4 | **39.0** | 49.8 | 58.4 |
| + NCA | **79.3** | 71.9 | **33.3** | 62.8 | 41.7 | **32.6** | 89.5 | **90.3** | 80.0 | 49.2 | 38.2 | 39.6 | **59.0** |
| Proprietary Models | | | | | | | | | | | | | |
| GPT-3.5 Turbo | 76.8 | 82.5 | 23.3 | 61.2 | 37.8 | 35.6 | 83.0 | 90.6 | 70.1 | 56.6 | 29.4 | 36.9 | 57.0 |
| GPT-4 | 85.4 | 83.5 | 41.8 | 85.6 | 69.7 | 52.4 | 94.8 | 92.6 | 86.7 | 79.7 | 59.6 | 65.8 | 74.8 |

2023). We finetune base models for 1 epoch with a 2e-5 learning rate and 0.1 warmup ratio using a cosine scheduler. For EURUS-7B, we mix 32K UltraChat, 30K ShareGPT, and 50K OpenOrca. For For EURUS-70B, we mix 63K UltraChat, 30K ShareGPT, and 70K OpenOrca.

**Perference Learning.** Based on EURUS-SFT models, we explore three preference learning algorithms, DPO (Rafailov et al., 2023), KTO (Ethayarajh et al., 2024), and NCA (Chen et al., 2024a). Differently from SFT, here we include all multi-turn trajectory pairs in our ULTRAINTERACT (220K) and include all UltraFeedback (Cui et al., 2023) pairs (340K). For hyperparameters, all $\beta$ is set to 0.1, and $\lambda_+/\lambda_-$ in KTO is set to 1.33 as recommended. We finetune models for 1 epoch with a 5e-7 learning rate and 0.1 warmup ratio using a cosine scheduler.

**Reward Modeling.** Similarly to the preference learning, we use all 220K multi-turn trajectory pairs from ULTRAINTERACT; it is further augmented with the 240K single-turn action pairs from ULTRAINTERACT. More details are in the Appendix B. We include all 340K pairs from UltraFeedback and one pair for each instruction from UltraSafety (Guo et al., 2024b), totaling 3K. EURUS-RM-7B is initialized from EURUS-7B-SFT with a new linear layer. We train RM for 1 epoch with lr=1e-5 learning rate. We also use a cosine scheduler with a warmup ratio of 0.1.

Our findings in §6 indicate that the absolute values of rewards make a big difference in the models' reasoning performance. We therefore augment the established Bradley-Terry (BT) objective $\mathcal{L}_{BT}$ with an additional term $\mathcal{L}_{DR}$ to **dir**ectly increase the reward of the chosen actions for instances from

ULTRAINTERACT, and decrease those of the rejected ones:

$$\mathcal{L}_{\text{ULTRAINTERACT}} = \underbrace{-\log\left(\sigma\left(r_\theta\left(x, y_c\right) - r_\theta\left(x, y_r\right)\right)\right)}_{\mathcal{L}_{\mathcal{BT}} \text{ optimize relative rewards}}$$

$$\underbrace{-\log\left(\sigma\left(r_\theta\left(x, y_c\right)\right) - \log\left(\sigma\left(-r_\theta\left(x, y_r\right)\right)\right)\right)}_{\mathcal{L}_{\mathcal{R}:} \text{ : increase } r_\theta(x, y_c) \text{ and decrease } r_\theta(x, y_r)}$$

For instances from other datasets, we train with $\mathcal{L}_{\text{BT}}$. $\theta$ denotes the reward model's parameters, $r_\theta\left(\cdot\right)$ and $r_\theta\left(x, y_r\right)$ the rewards on the chosen and rejected actions respectively. Our ablation study demonstrates the importance of both $\mathcal{L}_{\text{BT}}$ and $\mathcal{L}_{\text{DR}}$.

# 4. Evaluation of EURUS-7B and EURUS-70B

**Evaluation Setup.** We consider both single-turn and multi-turn reasoning. For single-turn evaluation, we consider HumanEval (Chen et al., 2021), MBPP (Austin et al., 2021), and LeetCode (Guo et al., 2024a) for coding, GSM-Plus (Li et al., 2024), MATH, TheoremQA (Chen et al., 2023), SVAMP (Patel et al., 2021), and ASDiv (Miao et al., 2020) for math, and BBH-Hard (Suzgun et al., 2022) for reasoning. We evaluate with pass@1 accuracy. We also use IFEval (Zhou et al., 2023) to assess the instruction-following ability and report the prompt-level loose score. For multi-turn evaluation, we adopt MINT (Wang et al., 2023b) and only consider the coding and math problems. We report the success rate at Turn 5. Please find further details on evaluation setups and evaluations beyond reasoning in Appendix C.

As shown in Table 2, we compare our EURUS with general-purpose models, and those specialized in coding and math of various sizes. We also summarize the results of GPT-3.5 Turbo and GPT-4 reported in previous works.

## 4.1. Results

Results are shown in Table 3. We summarize the takeaways as follows:

**EURUS, both the 7B and 70B variants, achieve the best overall performance among open-source models of similar sizes. EURUS even outperform specialized models in corresponding domains in many cases.** Notably, EURUS-7B outperforms baselines that are 5× larger and **EURUS-70B achieves better performance than GPT-3.5 Turbo**. EURUS's instruction-following performance is among the best general-purpose models, substantially better than specialized ones.

**Preference learning with ULTRAINTERACT can further improve the performance, especially in math and the multi-turn ability.** KTO and NCA consistently improve the models' performance in all five math benchmarks and multi-turn evaluations, while their effects vary in others. Since SFT models only use the single-turn data from ULTRAINTERACT while preference learning uses the multi-turn ones,

the improvements in interaction ability should also be attributed to ULTRAINTERACT rather than the algorithms alone. Surprisingly, we observe that **DPO hurts model performance on most benchmarks**. DPO training of our 70B model fails since the rewards go down to $-\infty$. We analyze this phenomenon in §6.1.

# 5. Evaluation of EURUS-RM-7B

**Evaluation Setup.** We evaluate EURUS-RM-7B on three RM benchmarks, RewardBench (Lambert et al., 2024), AutoJ (Li et al., 2023a), and MT-Bench (Zheng et al., 2023). Aiming for a more realistic OOD evalation, we exclude the "prior sets" split from RewardBench, since many baselines train on the datasets that this split contains. We compare with PairRM (Jiang et al., 2023b), Starling-RM-7B/34B (Zhu et al., 2023), UltraRM-13B (Cui et al., 2023), GPT-3.5 Turbo, and GPT-4. To further explore EURUS-RM-7B's potential in improving models' performance through reranking, we use it to rerank Mistral-7B-Instruct-v0.2's responses on HumanEval, MBPP, GSM8K, and MATH. We report the results of random sampling, self-consistency, and Starling-RM-34B as baselines.

## 5.1. Results

Table 4 summarizes reward modeling performance, and Figure 4 plots some reranking results with others in Appendix D.1.

**EURUS-RM-7B stands out as the best 7B RM overall, and achieves similar or better performance than much larger baselines. Particularly, it outperforms GPT-4 in certain tasks.** EURUS-RM-7B achieves a better correlation with human experts than all existing models on AutoJ and MT-Bench, and it achieves comparable performance to the 5× larger Starling-RM-34B on RewardBench. On RewardBench, EURUS-RM-7B outperforms all baselines on the "Chat-Hard" split while achieving very competitive performance on the "Reasoning" split. Across the AutoJ splits, EURUS-RM-7B outperforms nearly all existing models, with the only exception being GPT-4's results on Coding.

**Our training objective is beneficial in improving RM performance on hard problems and reasoning.** Table 4 shows that optimizing $\mathcal{L}_{\text{DR}}$ improves RM's reasoning ability, but BT modeling is still beneficial in equipping RM with abilities in general chatting as suggested in the "Chat-Hard" column, though its effect on reasoning may vary.

**ULTRAINTERACT is compatible with other datasets like UltraFeedback and UltraSafety, and mixing these datasets can balance different RM abilities.** Improving RM's capabilities in reasoning with ULTRAINTERACT does not sacrifice others, which indicates that ULTRAINTERACT can be a great ingredient for the training data mixture of

*Table 4.* Results on reward modeling benchmarks. UF: UltraFeedback; US: UltraSafety. The best performance in each benchmark is in bold and the second best one is underlined. Most baseline results are from (Jiang et al., 2023b) and (Lambert et al., 2024).

| Model | Reward Bench | | | | | AutoJ | | | | MT-Bench |
|---|---|---|---|---|---|---|---|---|---|---|
| | Chat | Chat-Hard | Safety | Reasoning | Avg. | Code | Math | Others | Overall | |
| PairRM | 90.2 | 53.0 | 31.5 | 60.0 | 58.7 | 58.3 | 52.8 | 58.9 | 59.1 | 59.0 |
| Starling-RM-7B | 98.0 | 43.4 | 88.6 | 74.6 | 76.2 | 59.2 | 47.2 | 61.4 | 60.8 | 56.8 |
| Starling-RM-34B | 96.9 | 59.0 | 89.9 | 90.3 | **84.0** | 65.8 | 54.2 | 62.3 | 62.6 | 60.4 |
| UltraRM-13B | 96.1 | 55.3 | 45.8 | 82.0 | 69.8 | 55.0 | 43.1 | 59.6 | 59.9 | 56.0 |
| GPT-3.5 Turbo | - | - | - | - | - | 36.6 | 40.3 | 41.2 | 42.7 | 57.1 |
| GPT -4 | - | - | - | - | - | 69.2 | 51.4 | 61.4 | 61.9 | 63.9 |
| EURUS-RM-7B | 96.5 | 65.3 | 80.7 | 87.0 | 82.4 | 87.5 | 82.5 | 78.0 | 80.7 | 79.4 |
| w/o $\mathcal{L}_{DR}$ | 96.4 | 59.9 | 79.5 | 77.5 | 78.3 | 83.8 | 82.5 | 78.9 | 80.7 | 79.3 |
| w/o $\mathcal{L}_{BT}$ | 96.8 | 58.5 | 83.8 | 84.2 | 80.8 | 88.8 | 92.5 | 79.4 | **81.9** | **79.6** |
| w/o US | 96.5 | 66.2 | 67.7 | 81.7 | 73.3 | 87.5 | 90.0 | 79.2 | 81.8 | 79.2 |
| w/o UF + US | 95.1 | 61.1 | 63.7 | 73.4 | 78.0 | 73.8 | 80.0 | 71.7 | 72.8 | 73.0 |

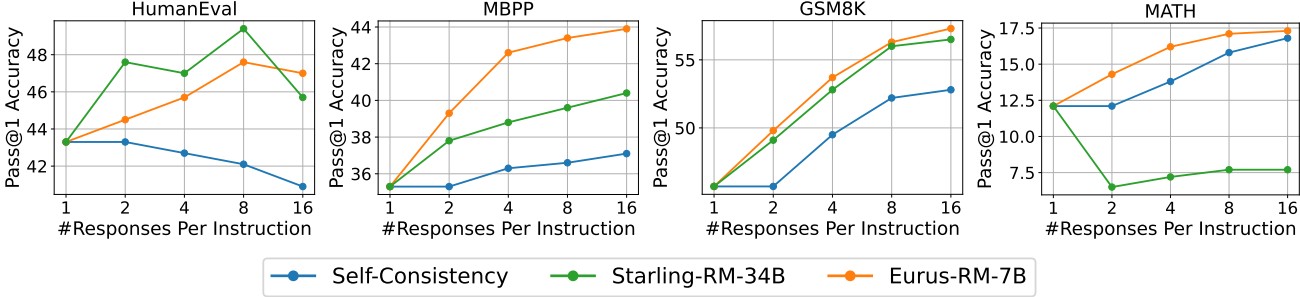

*Figure 4.* Results on reranking Mistral-7B-Instruct-v0.2's responses. Full results in Table 9.

reward models.

**EURUS-RM-7B improves LLMs' reasoning performance by a large margin through reranking.** EURUS-RM-7B consistently improves pass@1 accuracy across all tasks and performs better than 5× larger baseline Starling-RM-34B. Also, EURUS-RM-7B's reranking performance scales well with #responses per instruction, except a slight decrease in HumanEval when increasing response number form 8 to 16. In contrast, Starling-RM-34B suffers from severe performance drop on HumanEval and it consistently hurts model accuracy on MATH.

## 6. Analysis

### 6.1. Explicit Reward as A Proxy? Hypothesis for Preference Learning in Reasoning

We investigate the reason why DPO behaves differently than KTO and NCA. We start by empirically inspecting the rewards throughout the preference learning process, as shown in Figure 5. Rewards for chosen rejected data both keep decreasing through DPO, though the rewards for chosen data is still higher hence the loss decreases. In KTO and NCA, the rewards of chosen data keep increasing with those of rejected data decreasing.

Therefore, we hypothesize it is the distinction in the trend of rewards that leads to the performance gap between DPO and the other two algorithms. This distinction can be attributed to that DPO, derived from the Bradley-Terry model, only optimizes the relative differences between chosen and rejected data overlooking the absolute values of the rewards. This is a non-issue in alignment with general human values where preference is "relative" and there can be many valid answers to the same input. However, in reasoning tasks, the space of correct answers is much smaller than that of incorrect ones. Further, we notice that the rewards of chosen data in the last training step follow the ranking order of KTO > NCA > DPO, positively correlate with their performance trends. Therefore, we believe that increasing the rewards of the chosen data is especially beneficial in preference learning for reasoning tasks.

### 6.2. Ablation Study

We study the impact of ULTRAINTERACT and other open-source alignment data on EURUS-7B-SFT's performance. We consider three settings: (1) **With original ground-truth answers,** which replaces the generated actions with ground-truth rationales and answers from the original datasets. If no rationales are available, we use those from ULTRAINTERACT. (2) **Open-source data only.** (3) **ULTRAINTERACT**

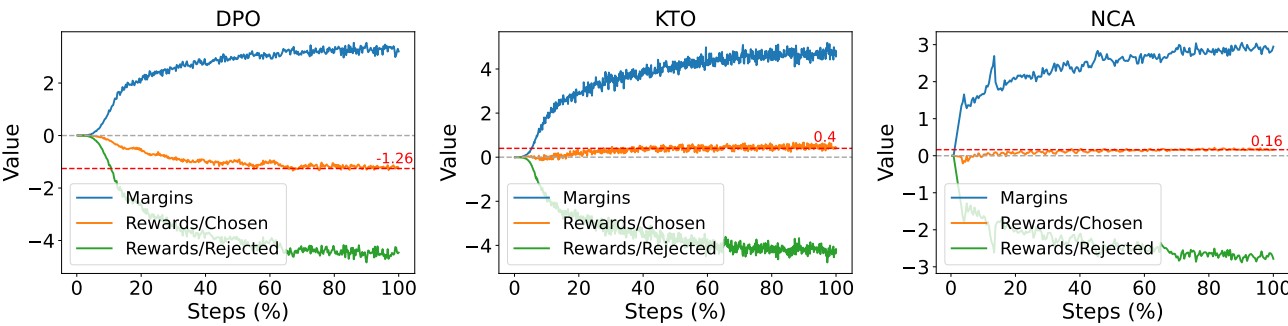

*Figure 5.* Reward patterns of EURUS-7B preference learning with DPO, KTO, and NCA. For all algorithms, the rewards of rejected data keep decreasing and the margins between chosen and rejected data keep increasing. However, the rewards of chosen data decrease below zero in DPO while keeping increasing and staying positive in KTO and NCA. The absolute values of the reward in the last step (in red) of the three algorithms positively correlate with their performance in Table 3.

*Table 5.* Ablation Study.

| Model | Coding | | | Math | | | | | Reasoning | Ins-Following | Avg. |
|---|---|---|---|---|---|---|---|---|---|---|---|
| | HumanEval | MBPP | LeetCode | GSM8K | MATH | TheoremQA | SVAMP | ASDiv | BBH | IFEval | |
| **EURUS-7B-SFT** | **55.5** | **59.1** | **20.0** | **73.7** | **32.6** | 20.0 | **82.2** | **84.1** | 64.6 | **44.0** | **53.6** |
| **Ground-Truth** | 46.3 | 46.4 | 8.9 | 62.2 | 15.0 | 9.6 | 75.1 | 68.8 | 64.4 | 42.9 | 44.0 |
| **Open-Source Only** | 38.4 | 44.1 | 11.1 | 45.3 | 10.8 | 9.3 | 52.7 | 49.4 | 65.3 | 43.6 | 37.0 |
| **ULTRAINTERACT Only** | 46.3 | 50.1 | 15.6 | 67.6 | 30.9 | **20.1** | 80.4 | 82.0 | **67.0** | 17.4 | 47.7 |

**only.** We evaluate with the same setting as §4.

In Table 5, EURUS outperforms the "Grouth-truth" model on all tasks, confirming the advantage of ULTRAINTERACT's designs of divide-and-conquer and code-as-action patterns, in line with conclusions of concurrent work (Chen et al., 2024b; Wang et al., 2024). Training only on open-source data without ULTRAINTERACT greatly hurts the reasoning performance, confirming the effectiveness of ULTRAIN-TERACT. Meanwhile, training only on ULTRAINTERACT suffers a performance drop except for BBH, especially in instruction following. We attribute the performance drop to a worse instruction-following ability. This suggests the necessity of mixing ULTRAINTERACT with other alignment data for better all-around supervised fine-tuning.

## 7. Related Work

**Open LLMs in Reasoning.** Open-source LLMs have shown remarkable progress in building *specialists* that excel in mathematics reasoning (Luo et al., 2023a; Yue et al., 2023; Toshniwal et al., 2024) or coding abilities (Roziere et al., 2023; Wei et al., 2023; Guo et al., 2024a; Zheng et al., 2024). On the contrary, mastering general reasoning capabilities still challenges open models, while the most advanced ones (DeepSeek-AI, 2024; Bai et al., 2023; Touvron et al., 2023; Jiang et al., 2024) are well behind proprietary models. More, these cutting-edge open general-purpose models maintain their alignment recipes confidential, which further hinders the replication and development of open-source

reasoning models.

**Preference Learning for Reasoning.** Aligning language models from human or AI preferences has emerged as a prevalent approach in the open-source community (Tunstall et al., 2023; Bai et al., 2023) with the proposal of DPO (Rafailov et al., 2023) and high-quality preference datasets (Cui et al., 2023; Zhu et al., 2023). Different from open-domain chatbots, preference learning is largely underexplored in complex reasoning. Recent research showed performance degradation when applying DPO on reasoning tasks, but some newly proposed algorithms demonstrated a positive effect (Ethayarajh et al., 2024; Chen et al., 2024a; Mitra et al., 2024; Shao et al., 2024). However, a deep understanding of preference learning, specifically its efficacy on complex reasoning, is not yet established.

## 8. Conclusion

We strive to narrow the huge gap between open-source models and proprietary models from the perspective of alignment. Our work pushes the boundaries of open-source reasoning generalists by (1) releasing a high-quality multi-turn reasoning dataset ULTRAINTERACT with preference trees, (2) introducing EURUS-series LLMs which achieve new SOTA on challenging reasoning benchmarks and (3) providing insights on preference learning for reasoning through analysis, leading to new reward modeling objectives as well as a powerful reward model for reasoning.

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

# A. Additional Details in ULTRAINTERACT Construction

## A.1. Dataset Details

**Math.** We adopt **GSM8K** (Cobbe et al., 2021), **MATH** (Hendrycks et al., 2021b), **MathQA** (Amini et al., 2019), and **NumGLUE** (Mishra et al., 2022)for mathematic reasoning, and include **TabMWP** (Lu et al., 2023) for tabular processing. We retain all the instructions for all datasets except MathQA, NumGLUE, and TabMWP. MathQA divides problems into different categories according to the topics and annotates the formula that indicates the pattern needed to solve each problem. We apply stratified sampling to sample at most five problems for each pattern and prioritize the problems that come from the long-tail category. Numglue contains eight different reasoning tasks and we discard Task 5 (Reading Comprehension + Explicit Numerical Reasoning), Task 6 (Reading Comprehension + Implicit Numerical Reasoning), and Task 7 (Quantitative NLI) due to the simplicity (Mishra et al., 2022). For TabMWP, we only keep the questions with difficulty levels 4 and 5 since the rest are too easy for current state-of-the-art models.

**Code.** We focus on programming with Python for the simplicity of integration of the interpreter. We use **CodeContest** (Li et al., 2022) and **TACO** (Li et al., 2023b), two competition-level coding datasets collected from various online platforms. We filter out the overlapped questions. Note that part of the questions in TACO only contain ground-truth solutions and do not contain test cases for evaluation, hence we apply GPT-4 to generate 12 test case inputs (4 basic inputs, 4 edge cases, and 4 large numbers) for each question and then execute the ground-truth solution snippets to produce outputs. Given that the two datasets mainly focus on competition problems that may deviate from real-world daily uses, we exclusively adopt **Magicoder-Evol-Instruct** (Luo et al., 2023b; Wei et al., 2023), the only dataset in our selection that does not contain test cases or ground-truth solutions. We employ GPT-4 Turbo to judge the correctness of generated code during interaction, and therefore we do not use this dataset for preference learning since we cannot rigorously construct pairs of correct and incorrect actions limited by the evaluation reliability. We also include **WikiTableQuestions** (Pasupat & Liang, 2015) for table processing with code.

**Logical Reasoning.** we use the multi-hop reasoning datasets **HotpotQA** (Yang et al., 2018) and **StrategyQA** (Geva et al., 2021), and the logical reasoning dataset **ReClor** (Yu et al., 2020). We follow the setting of Wang et al. (2023b) and convert HotpotQA to a generation task, removing the contexts and requiring LLMs to search relevant information using Wikipedia API.

*Table 6.* ULTRAINTERACT covers a diverse set of datasets spanning three tasks.

| Task | Datasets |
|------|----------|
| **Math** | GSM8K (Cobbe et al., 2021), MATH (Hendrycks et al., 2021b), MathQA (Amini et al., 2019), NumGlue (Mishra et al., 2022), TabMWP (Lu et al., 2023) |
| **Coding** | CodeContest (Li et al., 2022), TACO (Li et al., 2023b), WikiTableQuestions (Pasupat & Liang, 2015), Magicoder-Evol-Instruct (Luo et al., 2023b; Wei et al., 2023) |
| **Logic** | ReClor (Yu et al., 2020), HotpotQA (Yang et al., 2018), StrategyQA (Geva et al., 2021) |

### A.2. Details on Preference Tree Construction

**Models Adopted for Incorrect Action Sampling.** We randomly sample one model from Mistral-7B-Instruct-v0.2, DeepSeek-Coder-33B-Instruct, Mixtral-8x7B-Instruct, and DeepSeek-LLM-67B-Chat to generate one incorrect action to pair with each correct one.

**Correct Action Generation Based on Ground Truth Annotations.**

We adopt GPT-3.5 Turbo as the generator to generate correct actions based on ground truth considering the instruction-following ability. We provide different access to the ground truth information for different tasks, specifically: (1) For coding, where test cases are black boxes to reference solutions, we provide full access to the solution codes. The actor model will add step marks and corresponding explanations to the ground-truth code to make it easier to understand, or further refine the code for optimization. (2) For tool-free math problems, to avoid the actor model directly copying the answers to pass the correctness checking, we mask the answer numbers in the rationale before providing it to LLMs. This approach can better ensure response quality since it encourages LLMs to generate responses with complete reasoning chains with each step clearly marked. (3) For program-enhanced math reasoning, we first translate the textual rationale into code. Then, we either directly provide it to the actor model to generate plans, or ask the actor model to convert the code into modularization programming and then make plans to create tools to solve problems.

### A.3. Data Decomtamination

We conduct careful decontamination. Firstly, for LeetCode, we apply the Exact Substring Matching Algorithm[2] to compare with each instruction in the ULTRAINTERACT and find no overlaps. For others, we perform 8-gram exact matching to compare ULTRAINTERACT instructions with test sets of the same task. We remove those instructions that overlap 8 grams with any test sample.

[2]https://github.com/bigcode-project/bigcode-dataset/tree/main/decontamination

### A.4. Detailed Statistics

In total, ULTRAINTERACT has 86K instructions and 220K action pairs. The Total # Pairs does not equal Total # Turns in ULTRAINTERACT, since we fail to generate sufficient correct actions for every incorrect action in multi-turn trajectories mainly due to a lack of sufficient ground truth annotations. The total # pairs may not equal # correct answers, either, because it is also difficult and unnecessary to sample incorrect actions for the correct ones for some simple instructions. We present the specific information for each dataset. In particular, we list information on human annotation in each dataset, which plays an important role in correct action generation (§2.3 and Appendix A.2). All three steps of correct action sampling methods mentioned in §2.3 can be applied to datasets that have rationales, while for datasets only containing answers, only the first two steps are applicable. We do not apply any of the three-step methods to generate correct answers for Magicoder, the only dataset without any human annotation, to construct preference pairs.

## B. Additional Details on Training EURUS Models

Regarding pair augmentation, we scale up the pairs by matching every correct action for each instruction with one incorrect action of other turns. This leads to NxN pairs of single-turn actions for a trajectory of depth N. We remove the action pairs consisting of nodes at the same turn, as they are already part of the multi-turn trajectory pairs we included. Next, to avoid overfitting on the training set, we only select instructions with NxN ≤ 10, and for these instructions, we randomly sample at most 9 pairs with each action occurring no more than 3 times. This leads to an augmentation of 240k single-turn action pairs.

## C. Additional Evaluation Results of EURUS

**Detailed Setup in §4.** For math, we test both textual reasoning and program-enhanced settings and report the best performance of the two. All evaluations are conducted in 0-shot CoT with two exceptions: BBH uses 3 shots and IFEval does not use CoT. For MINT, we select MATH, TheoremQA, and MMLU-math from "reasoning" as a new

*Table 7.* Stats breakdown

| Task | Dataset | w/ Tool? | # Prompts | # Pairs | # Correct Answers. | Avg. Length | Human Annotation | |
|---|---|---|---|---|---|---|---|---|
| | | | | | | | Has Answer? | Has Rationale? |
| Math | GSM8K | ✓ | 4,522 | 10,277 | 17,392 | 1,746.7 | ✓ | ✓ |
| | | ✗ | 7,257 | 10,879 | 15,752 | 823.3 | ✓ | ✓ |
| | MATH | ✓ | 7,474 | 22,905 | 34,667 | 1,189.0 | ✓ | ✓ |
| | | ✗ | 7,471 | 25,765 | 36,005 | 1,735.0 | ✓ | ✓ |
| | MathQA | ✓ | 7,552 | 15,079 | 20,328 | 2,338.5 | ✓ | ✓ |
| | | ✗ | 7,159 | 17,743 | 22,500 | 1,916.3 | ✓ | ✓ |
| | NumGLUE | ✓ | 3,020 | 3,601 | 5,717 | 1,474.6 | ✓ | ✗ |
| | | ✗ | 2,835 | 2,987 | 4,273 | 1,056.1 | ✓ | ✗ |
| | TabMWP | ✓ | 3,117 | 4,135 | 6,083 | 842.6 | ✓ | ✗ |
| Coding | CodeContest | - | 8,167 | 44,319 | 44,666 | 2,061.7 | ✓ | ✓ |
| | TACO | - | 9,016 | 50,877 | 58,191 | 2,143.5 | ✓ | ✓ |
| | WikiTableQuestions | - | 1,401 | 1,544 | 1,738 | 1,794.8 | ✓ | ✗ |
| | Magicoder-Evol-Instruct | - | 10,374 | 0 | 10,238 | 687.1 | ✗ | ✗ |
| Logic | Reclor | ✗ | 4,467 | 7,958 | 7,231 | 1,266.7 | ✓ | ✗ |
| | HotpotQA | ✓ | 1,182 | 1,009 | 1,230 | 1,333.2 | ✓ | ✗ |
| | StrategyQA | ✓ | 904 | 741 | 968 | 1,256.2 | ✓ | ✗ |

*Table 8.* MMLU and MT-Bench.

| Model | MMLU | MT-Bench |
|---|---|---|
| ~7B | | |
| Mistral-7B-Instruct-v0.2 | 58.9 | 7.60 |
| Zephyr-7B-$\beta$ | 59.7 | 7.34 |
| OpenChat-3.5-1210 | 63.4 | 7.81 |
| Starling-LM-7B-$\alpha$ | **64.0** | **8.09** |
| Magicoder-S-DS-6.7B | 37.1 | - |
| OpenCI-DS-6.7B | 37.2 | - |
| MAmmoTH-7B-Mistral | 56.2 | - |
| WizardMath-7B-v1.1 | 60.3 | - |
| OpenMath-Mistral-7B | 58.3 | - |
| EURUS-7B-SFT | 61.8 | 7.15 |
| + DPO | 62.4 | 7.38 |
| + KTO | 62.2 | 7.38 |
| + NCA | 62.2 | 7.38 |
| ~40B | | |
| Mixtral-8x7B-Instruct | **70.3** | **8.30** |
| DeepSeek-Coder-33B-Ins | 40.2 | - |
| ~70B | | |
| CodeLLaMA-70B-Instruct | 55.1 | - |
| DeepSeek-LM-67B-Chat | 72.3 | - |
| QWen1.5-72B-Chat | **72.9** | **8.61** |
| OpenCI-CL-70B | 52.4 | - |
| OpenMath-CL-70B | 60.2 | - |
| EURUS-70B-SFT | 59.1 | 7.69 |
| + KTO | 59.5 | 7.93 |
| + NCA | 59.4 | 7.54 |
| Proprietary Models | | |
| GPT-3.5 Turbo | 70.0 | 7.94 |
| GPT-4 | **86.4** | **8.96** |

"math" split. We also evaluate 5-shot MMLU (Hendrycks et al., 2021a) for STEM knowledge and MT-Bench (Zheng et al., 2023) for conversation abilities to study whether EURUS needs to trade off other capabilities for reasoning.

**Results.** Results are shown in Table 8.

On MMLU, EURUS outperforms baselines dedicated to coding and math, and achieves higher results than Mistral-Instruct-v0.2 and CodeLLaMA-70B-Instruct, the official aligned versions of our base model built by their authors. Compared to general-purpose baseline models, EURUS-7B achieves comparable performance with the top-performance OpenChat and Starling-LM, though EURUS-70B does not achieve the same level of performance as other general-purpose models, which is expected due to the gap in the base models since CodeLLaMA-70B has not been intentionally optimized for knowledge.

On MT-Bench, we report baseline numbers from the official leaderboard[3]. EURUS matches the performance of mainstream open-source general-purpose models, and EURUS-70B-KTO further achieves the score of GPT-3.5 Turbo.

# D. Detailed Results on Reward Modeling

## D.1. Additional Results on Reranking

We present the full results on reranking in Table 9, where the conclusions are consistent with those drawn from §D: (1) Our reward models always achieve the highest accuracy on all test sets across different N, except when N=2

---

[3] https://huggingface.co/spaces/lmsys/chatbot-arena-leaderboard

*Table 9.* Detailed results of reranking Mistral-Instruct-v0.2's responses on coding and math.

| Datasets | HumanEval | | | | MBPP | | | | GSM8K | | | | MATH | | | |
|---|---|---|---|---|---|---|---|---|---|---|---|---|---|---|---|---|
| N | 2 | 4 | 8 | 16 | 2 | 4 | 8 | 16 | 2 | 4 | 8 | 16 | 2 | 4 | 8 | 16 |
| Random | 41.5 | 39.0 | 40.2 | 39.6 | 33.1 | 33.6 | 34.3 | 30.1 | 45.0 | 43.1 | 44.5 | 40.2 | 11.5 | 11.3 | 10.0 | 8.5 |
| Top Logits | 43.3 | 43.3 | 43.3 | 43.3 | 35.3 | 35.3 | 35.3 | 35.3 | 45.7 | 45.7 | 45.7 | 45.7 | 12.1 | 12.1 | 12.1 | 12.1 |
| Self-Consistency | 43.3 | 42.7 | 42.1 | 40.9 | 35.3 | 36.3 | 36.6 | 37.1 | 45.7 | 49.5 | 52.2 | 52.8 | 12.1 | 13.8 | 15.8 | 16.8 |
| Starling-RM-34B | **47.6** | **47.0** | **49.4** | 45.7 | 37.8 | 38.8 | 39.6 | 40.4 | 49.1 | 52.8 | 56.0 | 56.5 | 6.5 | 7.2 | 7.7 | 7.7 |
| EURUS-RM-7B | 44.5 | 45.7 | 47.6 | 47.0 | 39.3 | **42.6** | **43.4** | **43.9** | **49.8** | 53.7 | 56.3 | 57.3 | 14.3 | 16.2 | 17.1 | 17.3 |
| w/o $\mathcal{L}_{\mathcal{DR}}$ | 45.7 | 44.5 | 46.3 | 50.0 | 39.3 | 42.4 | 42.4 | 42.1 | 49.4 | 53.2 | 55.4 | 56.3 | 14.2 | 16.1 | 17.0 | 16.9 |
| w/o $\mathcal{L}_{\mathcal{BT}}$ | 45.1 | 44.5 | 47.0 | 48.2 | 38.6 | 40.6 | 39.6 | 40.1 | 49.1 | 52.5 | 55.2 | 57.8 | 14.3 | 16.3 | 17.2 | 17.1 |
| w/o US | 45.7 | **47.0** | **49.4** | 50.6 | **39.3** | 41.1 | 41.4 | 42.9 | 49.4 | **53.8** | **57.4** | **58.7** | **14.5** | **16.6** | 17.2 | **17.5** |
| w/o UF + US | 43.9 | 43.3 | 47.0 | 46.3 | 36.3 | 38.1 | 36.6 | 35.3 | 49.4 | 52.3 | 54.6 | 57.2 | 14.3 | 16.5 | **17.4** | 17.4 |
| Pass@N | 62.8 | 73.8 | 88.4 | 92.7 | 42.4 | 48.1 | 52.6 | 58.6 | 54.9 | 64.1 | 73.2 | 80.4 | 16.9 | 22.7 | 28.9 | 35.5 |

on HumanEval. (2) Both $\mathcal{L}_{BT}$ and $\mathcal{L}_{DR}$ consistently help improve reranking performance on three test sets except for HumanEval, where removing either of the objectives can prevent the accuracy from dropping when increasing N from 8 to 16. (3) Modeling safety hurts reranking performance in reasoning. When removing UltraSafety from the training data, the RM achieves higher accuracies than EURUS-RM-7B except on MBPP.

