# OpenReview forum: "Advancing LLM Reasoning Generalists with Preference Trees"
_ICML.cc/2024/Workshop/AI4MATH — ICML 2024 Workshop AI4MATH Poster_

### Official Review · Reviewer_veXk · 2024-06-11

**Rating:** 6
**Confidence:** 3

**Summary:**

This paper introduces a dataset called ULTRAINTERACT for alignment with respect to reasoning tasks. Compared to prior such datasets, it includes more fine-grained information in the form of unbalanced binary trees. They finetune a 7B and 70B model from Mistral, CodeLlama respectively on their new dataset showing improved performance for reward modelling, further bolstered by a new reward objective they introduce.

**Questions:**

1. Could you compare with Llama-3 8B/70B as well to get the comparison with most up-to-date SOTA models?
2. You mentioned that you choose tasks which are difficult for GPT-3.5. I am curious why not choose tasks which are instead difficult for GPT-4? Is this because the critique model is GPT-4?
3. Could you report the overall expenses of interacting with the OpenAI API for producing ULTRAINTERACT? I think this information can be beneficial for further community efforts in producing such datasets.
4. What are some reasons that you expect make Eurus better on TheoremQA compared to other models, but not for other datasets?

Suggestions:
1. There is a typo in appendix section A.3: "Decomtamination" -> "Decontamination"
2. On line ~150, include the period . inside the quotations “environment”. -> "environment."

**Reasons To Accept:**

1. The ULTRAINTERACT dataset and other artifacts should be useful for the research community.
2. For reward modelling, the results indicate to be generally positive for Eurus compared to prior approaches of similar size, even compared to larger models like GPT-3.5 and GPT-4.
3. The analysis of different reward objective performance can be useful for further research.

**Reasons To Reject:**

1. Table 3 indicates that the Eurus models are not uniformly better than previous models, and there are some tasks it performs exceptionally better than, but suffers in others. The variance is most clear in the 7B comparison, likely less in the 70B comparison because there are not many quality models of that size which are open-sourced. However, I think the main claim in the paper is the performance on reward modelling.

---

### Official Review · Reviewer_QhqV · 2024-06-12

**Rating:** 5
**Confidence:** 3

**Summary:**

The paper presents an instruction finetuning and preference learning dataset for reasoning tasks like mathematics and coding. The dataset has been generated with GPT-3.5 and GPT-4. It contains positive and negative samples that employ tool use in the form of a Python interpreter and has 18 % multi-turn interactions with program execution feedback and critique feedback. Multi-turn samples come with correct and incorrect actions at each turn. By finetuning publicly available LLMs on this dataset, the authors show strong performance with respect to similarly sized models on reasoning benchmarks. Different preference learning methods are compared (DPO, KTO, NCA). An alternative reward modeling loss is suggested and a reward model trained on this dataset achieves strong performance on reward modeling benchmarks, especially for reasoning tasks.

**Questions:**

- Line 269: “Warmup ratio of 0.1” means 0.1 epochs?
- Which evaluation parameters were used? Greedy decoding?
- Which total batch sizes were used for the finetunings? (This is an important hyperparameter in finetunings on small dataset

**Reasons To Accept:**

- Sequence distillation is a relevant method for adjusting smaller LLMs to specific use cases in practice. The paper’s contribution lies in suggesting an improved method for extracting reasoning capabilities from larger models that are likely to be trained on more human-annotated data than the comparison models. The paper gives careful recipes to balance general chat capabilities and reasoning capabilities during instruction finetuning, notably data mixes and preference training algorithms.
- Different preference learning algorithms are compared in the specific domain of finetuning for reasoning. The observation that DPO turned out to be unstable and has negative reward estimates is interesting. Moreover, the authors suggest a new reward modeling objective.
- The paper has a rich empirical section with many evaluations and comparisons to competitor models on a variety of benchmarks and several ablative settings.

**Reasons To Reject:**

- Important ablations are missing that impact the message of the study.
  - The paper suggests that the dataset is better than comparable datasets because of its use of multi-turn evaluations and the availability of positive and negative actions for every turn. However, there is no comparison to a dataset generated with the same amount of computation / cost that does not use this “tree structure”. It could also be due to its use of (a recent version of) GPT-4 for difficult prompts and critique steps instead of older models, the choice of prompt sources, the inclusion of execution feedback etc. Apart from “use this dataset”, there is limited information to be gained for practitioners intending to produce their own finetuning datasets for reasoning.
  - In a similar vein, there are not nearly enough experiments to support the claims on the general usefulness of different finetuning methods (DPO, KTO, NCA) for preference learning. Each method could necessitate different optimal hyperparameters (learning rate, batch size, $\beta$, number of epochs), and before a general claim comparing the methods can be made, these choices should be ablated carefully. (And this is feasible: finetuning Mistral-7B is cheaper than generating from GPT-4; definite knowledge on methods can be more important than releasing a 70B model that is soon outdated anyway.) Training 70B models is known to be somewhat unstable, and a single failed DPO run is not enough to prove that the method generally fails.
- The analysis on DPO’s failure is incomplete: according to the DPO gradient (“What does the DPO update do?” in the original paper), the likelihood of correct answers is indeed increased while the likelihood of wrong answers is decreased. If therefore the authors observe (implicit) rewards of correct solutions to degrade, this means that the weight update cannot sufficiently distinguish between the behaviors that led to the good and bad outcomes, respectively, and fails at generalizing. Such observations are possible if there are imbalances or correlations in the data, possibly related to the way preference samples are extracted from multi-turn trajectories.
- The proposed alternative reward modeling objective is neither derived/explained nor ablated against different ways of incorporating “absolute terms” into the loss. The proposal amounts to augmenting each preference sample $(c, r)$ (correct, rejected) to three samples $(c, r), (c, 0), (0, r)$, where $0$ stands for a virtual sample with zero reward. Does this correspond to a baselined version of the Bradley-Terry model? Are there alternative ways of incorporating absolute terms into the reward modeling loss?
- The experimental sections should discuss advantages and disadvantages of the method more carefully. For instance, all preference learning methods appear to (at best) maintain / (at worst) severely harm the coding abilities of the 7B models.
- Evaluations could benefit from more samples to decrease variance, all numbers appear to be derived from a single sample per prompt.

---

### Official Review · Reviewer_UjRx · 2024-06-13

**Rating:** 8
**Confidence:** 3

**Summary:**

This paper introduces EURUS, a suite of LLMs optimized for reasoning, and proposes their training set, ULTRAINTERACT, for both SFT and Reward Modeling. ULTRAINTERACT is characterized by preference trees, diverse planning strategies, multi-turn interaction trajectories with the environment, and critique. By comparing and analyzing popular alignment approaches such as DPO, KTO, and NCA, the authors conclude with key findings in the alignment of reasoning tasks.

**Questions:**

1. How should the linear layer of a reward model be structured? Should it use the activation of the last token as input to the linear layer, or should it average the activations of all tokens?

**Reasons To Accept:**

1. Such multi-turn interaction augmented preference data, particularly focusing on reasoning tasks, is truly necessary for our community to develop agents with better reasoning abilities.
2. The paper is structured clearly, and the key points are well articulated.
3. The experimental results are quite extensive, and the final analysis is both intuitive and impressive.

**Reasons To Reject:**

1. It would be wonderful to see the scaling curves or experiments from EURUS-RM-7B to EURUS-RM-70B on ULTRAINTERACT, using the assistance from the policy model for selection.
2. According to the Figure 5, I find this figure very interesting, and you effectively demonstrate why DPO negatively impacts performance. I would like to ask if there is any way to improve DPO to avoid this issue, or which part of DPO causes this phenomenon? Could online DPO resolve this problem and enhance reasoning ability?

---

### Meta-Review · Area_Chair_JTCL · 2024-06-13

**Recommendation:** Accept (Poster)
**Confidence:** 4

**Metareview:**

Good paper. It proposes an instruction finetuning and preference learning dataset for reasoning tasks. The dataset contains multi-turn interactions with program execution feedback and critique feedback, which is novel to see. The paper also has a rich empirical section with many evaluations and comparisons to competitor models. Reviewers QhqV and veXk have made some solid points on the weaknesses of this paper, but overall, the contribution of this paper is significant and suitable to present in the workshop.

---

### Decision · Program_Chairs · 2024-06-13

Accept (Poster)